# Controlled Porosity of Selective Laser Melting-Produced Thermal Pipes: Experimental Analysis and Machine Learning Approach for Pore Recognition on Pipes Surfaces

**DOI:** 10.3390/s24154959

**Published:** 2024-07-31

**Authors:** Ivan Malashin, Dmitry Martysyuk, Vadim Tynchenko, Vladimir Nelyub, Aleksei Borodulin, Andrei Gantimurov, Anton Nisan, Nikolay Novozhilov, Viatcheslav Zelentsov, Aleksey Filimonov, Andrey Galinovsky

**Affiliations:** 1Artificial Intelligence Technology Scientific and Education Center, Bauman Moscow State Technical University, 105005 Moscow, Russia; dmart9945@mail.ru (D.M.);; 2Scientific Department, Far Eastern Federal University, 690922 Vladivostok, Russia; 3Engineering Center “Forta”, 117036 Moscow, Russia

**Keywords:** porosity, selective laser melting, additive manufacturing thermal pipes

## Abstract

This study investigates the methods for controlling porosity in thermal pipes manufactured using selective laser melting (SLM) technology. Experiments conducted include water permeability tests and surface roughness measurements, which are complemented by SEM image ML-based analysis for pore recognition. The results elucidate the impact of SLM printing parameters on water permeability. Specifically, an increase in hatch and point distances leads to a linear rise in permeability, while higher laser power diminishes permeability. Using machine learning (ML) techniques, precise pore identification on SEM images depicting surface microstructures of the samples is achieved. The average percentage of the surface area containing detected pores for microstructure samples printed with laser parameters (laser power (W) _ hatch distance (µm) _ point distance (µm)) 175_ 80_80 was found to be 5.2%, while for 225_120_120, it was 4.2%, and for 275_160_160, it was 3.8%. Pore recognition was conducted using the Haar feature-based method, and the optimal patch size was determined to be 36 pixels on monochrome images of microstructures with a magnification of 33×, which were acquired using a Leica S9 D microscope.

## 1. Introduction

In recent years, the application of selective laser melting (SLM) [1,2,3,4,5] technology has extended beyond traditional manufacturing sectors to encompass diverse fields, including thermal management systems such as heat pipes [6,7,8]. Heat pipes, due to their high thermal conductivity [9] and efficiency in heat transfer, are utilized in various engineering applications, ranging from electronics cooling [10] to aerospace propulsion [11] systems. Leveraging the capabilities of SLM, researchers have explored novel designs and configurations of heat pipes to enhance their performance and tailor them to specific applications.

The investigation of heat pipes fabricated using SLM involves a multidisciplinary approach, integrating techniques such as analysis of water permeability as a function of printing parameters [12,13,14], surface roughness characterization, and automated pore detection on the end faces of these pipes through microstructure image analysis. The goal is to optimize the analysis and manufacturing process of heat pipes through SLM while ensuring superior thermal performance and structural integrity by studying these aspects. Through this research, insights into the relationship between the printing parameters, surface quality, and functional properties of SLM-produced heat pipes can be gained. Furthermore, the aim is to employ ML techniques to determine the percentage of pores present on the external surface of these pipes.

Despite the extensive research on surface roughness in the scientific literature concerning SLM samples, the application of ML for pore recognition to determine the average percentage of pore surface remains limited. While surface roughness analysis provides valuable insights into the quality of SLM components, the automated detection and characterization of pores present on these surfaces pose significant challenges. Factors such as variability in pore morphology [3,15,16], size, and distribution, compounded by surface irregularities caused by the SLM process, contribute to the complexity of accurate pore recognition. As a result, there is a gap in the literature regarding robust methodologies for automated pore detection [17] and characterization on SLM surfaces using ML techniques.

Motivating this study is the need for precise and reliable methods to identify and quantify pores in SLM-produced heat pipes [18]. Pore recognition and analysis are essential for understanding and mitigating the effects of porosity on the structural integrity and thermal performance of these components. By leveraging ML techniques for pore detection, this research aims to provide a detailed characterization of the pore surface structures [19,20], thereby enabling the optimization of SLM parameters when producing heat pipes. This endeavor not only addresses the current limitations in pore recognition methodologies but also paves the way for enhanced quality control measures in the additive manufacturing of thermal management systems.

The article follows this structure: In Section 2, relevant research in the field is discussed. Subsequently, Section 3.1 elaborates on the specimen fabrication process. Section 3.2 delves into experiments on water permeability, while Section 3.3 addresses surface roughness measurements. Moreover, Section 3.4 and Section 3.5 involve the analysis of SEM images for pore recognition. Section 4.1 presents findings on the influence of printing parameters on water permeability, while Section 4.2 details surface roughness results, and Section 4.3 elucidates pore recognition outcomes.

## 2. Related Works

The heat pipe is a sealed enclosure, the internal walls of which are coated with a capillary-porous structure [21,22,23] called a wick (Figure 1). The pipes are evacuated (down to $10^{-5}$–$10^{-4}$ mm Hg) and filled with a working fluid (e.g., water, ammonia); the wick is completely filled with liquid, while the remaining internal space is saturated with its vapor.

The principle of operation of the heat pipe is as follows. When heat is applied to the evaporator, the liquid evaporates from the surface of the wick [24], transferring heat to the vapor. The vapor pressure in the pipe increases, disrupting the dynamic equilibrium of the vapor–liquid system in the rest of the pipe, causing vapor condensation on the surface of the wick. During vapor condensation on the wick surface [25,26], heat is transferred to the liquid and removed in the condenser. Then, the liquid returns through the wick to the evaporator due to capillary forces.

The attractiveness of using heat pipes is due to their extremely high effective thermal conductivity [27,28], reaching tens of thousands of W/(m·K), whereas the thermal conductivity of copper is 390 W/(m·K), and that of aluminum is 230 W/(m·K).

The manufacture of pipes using SLM has been examined in various literature sources. Chen et al. [29] used metal AM to create oscillating heat pipes with SUS316L material. Experiments optimized printing parameters and assessed thermal performance. Inter-channel spacing reduced thermal interaction, enhancing oscillation effect and improving thermal performance. Tests showed a significant enhancement in equivalent thermal conductivity with spacing.

Thompson et al. [30] utilized SLM to create a compact flat-plate oscillating heat pipe (FP-OHP) with innovative features, including a Ti-6Al-4V casing and closed-loop mini-channel. Venting holes facilitated a unique depowdering process. Scanning Electron Microscopy (SEM) revealed channel wall characteristics crucial for heat transfer. The study also addressed SLM challenges such as surface quality and depowdering. The Ti-6Al-4V FP-OHP exhibited successful operation with an effective thermal conductivity of approximately 110 W/mK at 50 W power input, representing a 400–500% increase relative to solid Ti-6Al-4V.

Three-dimensional (3D)-printed wicks were used to study pore size effects on loop heat pipe (LHP) performance in [31]. These wicks, with controlled structural parameters, were fabricated to avoid the randomness and closed pores seen in traditional sintered wicks. The LHP, featuring a 3D-printed wick (d=200 μm) made of stainless steel, was successfully initiated in 100 s at 20 W and operated stably from 20 to 160 W, maintaining an allowable evaporator wall temperature of 100 °C. The minimum evaporator thermal resistance of the 3D-printed wick was 0.031 K/W at 140 W with a corresponding maximum heat transfer coefficient of 44,379 W/m^2^K.

Esarte et al. [26] explore SLM 3D printing technology for fabricating primary wicks in loop heat pipes (LHPs), aiming to achieve precise control over internal passage geometry for optimal performance. The study highlights the impact of fluid charge and ambient temperature on LHP operation. Additionally, a case study is presented involving an 80 W LED street lamp cooled by an LHP equipped with the optimized wick, emphasizing the importance of this optimization in achieving efficient performance under different operating conditions.

Liu et al. [32] analyzed gas fluidity in cellular structures fabricated by SLM using 17-4 PH stainless steel, Inconel 718, and Ti-6Al-4V alloys. Porosity and permeability were measured, showing a relation to energy density. Inconel 718 displayed higher air permeability due to smoother pore pathways, while Ti-6Al-4V had the highest specific surface areas.

Jafari et al. [33] investigate the thermal conductivity and wicking properties of a stainless steel 316L porous structure fabricated via SLM technology, which is relevant to applications like heat pipes (HPs). The rectangular sample, with 46.5% porosity and dimensions of 20 × 40 × 1 mm³, is saturated with distilled water and ethylene glycol for analysis. Experimental procedures for determining effective thermal conductivity are outlined and compared with literature correlations. The Washburn wicking model is utilized, and contact angles with three test liquids (n-hexane, water, and ethylene glycol) are measured. Results affirm SLM’s viability for HP technology, with effective thermal conductivity ranging from 1.8 to 6.0 W/m·K for various working fluids.

Monroe [34] investigated a Ti-6Al-4V flat-plate oscillating heat pipe fabricated via SLM, which was characterized by capillary-sized circular mini-channels. Post-SLM, the prototype underwent a novel depowdering method and was charged with acetone, achieving an 800% increase in effective thermal conductivity relative to pure Ti-6Al-4V. Inspection via scanning electron microscopy revealed a surface roughness of approximately 45 micrometers, demonstrating AM’s potential for custom heat transfer media.

Xie et al. [19] investigate how laser power in the SLM process affects porosity in ZK60 magnesium alloys. High-speed monitoring showed that lower power increases pore coverage, while higher power results in denser structures. SEM and OM analyses confirmed these findings, revealing smaller and fewer pores at higher laser power. XCT provided 3D insights, highlighting a preference for interconnected pores at lower power and more isolated pores at higher power. Managing these factors is crucial for optimizing the SLM process of Mg alloys to control porosity effectively.

Cerezo et al. [17] investigate the fatigue behavior of additive manufactured samples by examining the interaction between building parameters, particularly fabrication angles, and pore presence. The study aims to characterize these pores and understand their impact on material fatigue properties. Through systematic porosity analysis in different fabrication orientations and detailed examination using energy-dispersive X-ray spectroscopy, consistent behavioral patterns emerged. Variables such as pore area and aspect ratio were found to significantly influence sample behavior in fatigue testing. These findings are crucial for advancing academic research in material science and engineering.

Titanium alloy porous scaffolds, incorporating triply periodic minimal surface (TPMS) designs like Gyroid and Diamond units, mimic natural bone structures and mitigate stress-shielding effects. Ye et al. [35] highlighted structures designed via selective laser melting; these scaffolds enhance nutrient transport and withstand complex stress environments. This research underscores the scaffold’s ability to meet demanding structural strength and permeability criteria essential for tissue engineering applications.

Table 1 summarizes various studies exploring different aspects of additive manufacturing in heat pipes and related structures. It highlights the diverse applications of SLM technology, ranging from optimizing heat transfer properties to studying material behavior under various conditions.

A knowledge gap exists regarding the impact of SLM printing parameters on the water permeability of AlSi10Mg thermal pipes. Understanding this relationship is neccessery for optimizing thermal pipe design and performance in applications like heat exchangers and cooling systems.

## 3. Materials and Methods

### 3.1. Fabrication of Samples

There are at least two methods for obtaining porous structures by SLM. One approach involves printing thin lattice structures with small distances between lattice nodes (typically less than 1 mm), thereby achieving a coherent porosity with pore sizes of several hundred micrometers (Figure 2a). However, this method presents difficulties in removing unmelted powder from the lattice, especially when reducing the distance between lattice nodes. In this study, an alternative method was employed: key printing parameters (Table 2) were adjusted to promote powder sintering rather than melting, resulting in the formation of pores tens of micrometers in size (Figure 2b).

We utilized the SLM 280 HL [36,37] (SLM solutions, Lübeck, Germany) printing machine. The utilized powder of AlSi10Mg consisted predominantly of aluminum (approximately 90%), silicon (about 9%), and magnesium (approximately 1%) by weight. The average particle size of the powder was 20 micrometers (μm) with a distribution range of 10 μm. Figure 3a presents a detailed 3D model of the internal structure of the sample of fabricated tube, and all printed samples are shown together in Figure 3b. Each sample measures 6 cm in height (h) and 2 cm in diameter (d) with hole diameters approximately 0.5 cm.

The objective was to create a more manageable porosity structure by promoting powder sintering, which would facilitate powder removal while still providing the desired level of permeability. This shift in focus from lattice printing to powder sintering opens up new possibilities for controlling pore size and distribution in SLM-fabricated parts, offering improved flexibility and efficiency in the manufacturing process. Eighteen samples were printed and tested for every 3-factor condition in the permeability experiment, totalling 54.

### 3.2. Water Permeability

The objective of the experiment was to assess the influence of key melting process parameters on the permeability of the porous structure and to preliminarily determine the range of achievable permeability values. These parameters for controlling coherent porosity were selected as laser power, point distance, and hatch distance, as shown in Figure 4. A full factorial experiment was conducted: for each combination of the varied factors, cylindrical samples with a porous structure area of ⊘17×30 mm were printed, and the permeability *K* was experimentally determined on these samples according to Darcy’s law [38,39] of fluid filtration using the following formula:(1)$$K=\frac{Q\mu l}{\Delta PF},$$
where *Q* is the volumetric flow rate of fluid through the sample per unit time, m³/s, μ is the absolute viscosity of the fluid, Pa·s, *l* is the length of the sample, m, ΔP is the pressure drop across the opposite ends of the tested sample, Pa, and *F* is the cross-sectional area of the sample, m².

Samples with porous structures were printed on a single-laser 3D printer with a maximum laser power of 400 W and a focal spot diameter of approximately 80 μm from AlSi10Mg alloy produced by RUSAL (Moscow, Russia), and they underwent standard heat treatment: heating to 300 °C for 45 min, holding at 300 °C for 2 h, and air cooling.

The experimental setup, illustrated in Figure 5, involved pumping water through each sample with a pressure drop of 2.5 bar for 5 min. The mass of water passing through was measured, and permeability was calculated according to Darcy’s law. Twelve samples did not allow water to pass through, partial failure (washing out of powder from the end face) was observed in one sample during water flow, and the permeability values obtained for the remaining samples ranged from $8.3\times 10^{-16}$ to $7.1\times 10^{-13}$ m².

### 3.3. Roughness Analysis

Surface roughness analysis was conducted to evaluate its impact on water permeability on the same samples used in the permeability experiments. For the analysis of surface roughness, each specimen was subjected to scanning using an optical profilometer, specifically the Profilograph-Profilometer BV-7669M (manufactured by AO “NIIizmereniya”, Moscow, Russia). This instrument ensured high precision in capturing surface profiles, allowing for a detailed analysis of roughness parameters. Subsequent data processing, facilitated by specialized software, enabled the extraction of key roughness characteristics for further investigation of the surface properties of the specimens. Surface roughness analysis was conducted on 54 samples fabricated using SLM printing with varying printing parameters. The following parameters were recorded:Average Roughness ($R_{a}$): Higher $R_{a}$ values typically indicate a rougher surface, potentially leading to increased water droplet retention, which could enhance the water surface tension [40].Maximum Profile Peak Height ($R_{max}$): Higher $R_{max}$ values suggest the presence of taller peaks and deeper valleys on the surface [41], potentially creating more microscopic obstacles to water flow through pores and reducing water conductivity.Root Mean Square Roughness ($R_{q}$): $R_{q}$ also indicates surface roughness but considers the squares of deviations from the centerline [42], essentially measuring surface height variability. Higher $R_{q}$ values may indicate a rougher surface, potentially increasing water surface tension and reducing conductivity.Maximum Peak Height ($R_{p}$): Higher $R_{p}$ values indicate taller individual peaks on the surface, which can obstruct water flow and reduce conductivity.Maximum Valley Depth ($R_{v}$): Higher $R_{v}$ values indicate deeper valleys on the surface, which can reduce the effective contact area between water and the surface, potentially increasing water surface tension.Mean Profile Spacing ($S_{m}$): $S_{m}$ represents the average distance between profile elements. $S_{m}$ values indicate closer profile element spacing, potentially creating more intricate paths for water flow through pores and reducing conductivity.Vertex Density (S): Higher vertex density indicates a greater number of vertices per unit length of the profile, which can also reduce conductivity due to increased surface resistance.Percentage of Profile Elements Above the Mean Line ($tp_{50}$): A high percentage of profile elements above the mean line may indicate higher peaks on the surface, which can also reduce conductivity.

### 3.4. SEM Images Analysis

Microstructure images were acquired using a Leica S9 D microscope (Leica Microsystems, Wetzlar, Germany) [43] to identify optimal images for pore recognition. This was essential for permeability analysis, linking the area through which water can pass with the parameters used in the SLM printing process. Through the research, it was concluded that the most optimal photographs for this purpose were those captured at a magnification of 33 times. This conclusion was drawn after an analysis of various magnification levels (Figure 6) and their corresponding image qualities, taking into account factors such as resolution, clarity, and pore visibility. The 33× magnification level emerged as the most suitable balance between image resolution and pore visibility, providing the necessary level of detail for accurate pore recognition while minimizing image noise and distortion.

### 3.5. ML-Based Pore Recognition

The approach employs the Haar wavelet method [44,45,46], which entails breaking down an image into small regions known as “Haar-like features” and scrutinizing these features for pattern recognition purposes. Haar-like features are a method of feature extraction used in computer vision to detect texture patterns and objects within images. The fundamental principle involves rectangular filters of varying sizes and positions sliding across an image, computing differences in pixel intensities between adjacent regions.

In the context of pore recognition on monochrome images, Haar-like features [47,48,49] represent a distinct visual pattern, such as edges, corners, or texture variations, applied to highlight characteristics related to skin texture or material surface. These features may include areas with altered brightness or contrast, which are indicative of pores. This method is widely utilized in object detection and facial recognition algorithms for its effectiveness in capturing pertinent image attributes. To effectively use Haar-like features for pore recognition, it is necessary adjust parameters such as the size and position of the patches to optimize detection accuracy. To achieve this, we developed an image processing algorithm capable of automatically identifying and quantifying pores on sample surfaces. Figure 7 illustrates the schematic pipeline of the experiment.

The algorithm operates as follows: first, it analyzes and converts images of pore fragments to the CV_8U data type. Subsequently, it iterates through all pore images within the designated folder and conducts a comparison with the current fragment. If a sufficiently high degree of similarity is detected (exceeding the established matching threshold), the algorithm registers the presence of pores.

The matching threshold serves as a criterion for determining the adequacy of the match between the current image patch and the pore image loaded from the designated folder. This threshold enables us to ascertain when a sufficiently accurate match has been established.

By applying this algorithm to microstructure photographs obtained from samples produced under various printing parameters, it is possible to assess the influence of these parameters on pore formation. The detection and quantification of pores in these images facilitate the analysis of changes in porosity and their correlation with printing process parameters.

To assess the quality of the proposed method, we employed an approach based on comparing fragments identified by the program with pore fragments manually marked by an expert. Depending on the size of the patterns used for comparison with the Haar-like features, different numbers of recognized pores were obtained—sometimes there were false positives, and sometimes the necessary pores were missed. An attempt was made to determine the most appropriate pattern size [50] that would provide the most accurate results. To achieve this, an approach was applied akin to binary classification—calculating the ratio of the area of pores identified by the expert to the rest area, which served as true positives. Similarly, the ratio was calculated of the background area unrelated to pores in the image where pores were automatically detected to the background area where pores were manually identified, representing false negatives.

## 4. Results

### 4.1. Influence of Laser-Printing Parameters on Water Permeability

The linear effects and pairwise interactions, demonstrating the dependence of permeability on point distance, hatch distance, and laser power, are presented in Figure 8.

By varying the laser power (LP), hatch distance (HD), and point distance (PD), distinct trends in permeability were observed.

Firstly, concerning laser power, higher values led to lower permeability due to an enhanced fusion of powder particles, resulting in larger pores and smoother surface structures. Conversely, higher laser power settings exhibited lower permeability, which was attributed to inadequate fusion and the formation of smaller, less interconnected pores. Large pores are defined as regions with visible voids or irregularities larger than 150 micrometers (Figure 7 red squared areas). Conversely, fine pores are small and uniformly distributed with diameters less than 150 micrometers (Figure 7 green squared areas).

Secondly, small hatch distances resulted in denser structures with reduced pore size, limiting water flow and decreasing permeability. Conversely, larger hatch distances facilitated better fusion and larger pore formation, enhancing permeability.

Thirdly, point distance influenced permeability by affecting the density and distribution of pores. Decreasing point distance led to denser structures with smaller, more uniformly distributed pores, resulting in reduced permeability. Conversely, increasing point distance allowed for larger, less uniform pore formation, promoting higher permeability.

Moreover, interactions between these factors demonstrated synergistic or antagonistic effects on permeability. For instance, higher laser power combined with smaller hatch distances (HD × LP) resulted in increased permeability compared to similar power settings with larger hatch distances. Similarly, the interaction between hatch and point distances (HD × PD) showed increasing water permeability with increasinf laser power.

The laser power and scan speed are necessary for determining the heat input into the material. High laser power or slow scan speed increases the energy density [51], resulting in deeper and wider melt pools. This can lead to more complete melting of the powder particles and better fusion between layers [52], thereby reducing porosity. However, excessive energy input can cause keyhole formation and increased vaporization, potentially decreasing porosity. Conversely, low energy input may result in insufficient melting and weak interlayer bonding, also increasing porosity [53,54].

Moreover, the particle size distribution, shape, and packing density of the powder influence the initial porosity and the flow characteristics of the molten material [53]. Spherical particles with a narrow size distribution tend to pack more densely and melt uniformly [55], leading to lower porosity and smoother surfaces. Irregularly shaped particles or a broad size distribution can create voids and inconsistencies in the melt pool, resulting in higher porosity and rougher surfaces [53]. The permeability of the parts is directly related to the interconnected porosity within the material. As described, higher energy input can reduce porosity by ensuring better fusion of the powder particles, thereby decreasing permeability.

### 4.2. Roughness of Surfaces

Experiments were conducted by printing samples under different printing conditions to study the surface roughness of components produced using SLM, and the resulting surface roughness profiles were examined. Examples of surface roughness profiles for samples printed under different printing parameters are illustrated in Figure 9.

The distribution of surface roughness parameters $R_{a}$, $R_{\max}$, $R_{q}$, $R_{p}$, $R_{v}$, $S_{m}$, *S*, and $tp_{50}$, depending on the printing regime (Table 2), are shown on Figure 10. Printing regime refers to the specific set of parameters and conditions under which a print job is carried out. It displays histograms illustrating the distribution of these roughness parameters for the 54 samples printed under three different printing regimes. It is observed that when the laser power is set to 275 watts and the hatch and point distances are 160 μm, the surface roughness values often reach higher levels compared to the other two printing regimes. These histograms were generated based on the same 54 samples depicted in Figure 3b with 18 samples printed under each printing regime.

Furthermore, a correlation matrix (Figure 11) was constructed to examine the relationship between the SLM printing parameters and surface roughness parameters. Surprisingly, the results indicate that there is no significant influence of the printing parameters on the surface roughness. Potential reasons for this phenomenon could be attributed to factors such as the inherent characteristics of the SLM process, the material properties, and the interaction between laser energy and powder particles during printing. These factors may overshadow the effects of individual printing parameters on surface roughness, leading to the observed trends in the correlation matrix.

Several steps were undertaken to ensure the reliability and consistency of the roughness measurements. Firstly, the meticulous calibration [56] of a profilometer was conducted prior to surface roughness assessment. This calibration ensured precise and consistent measurement readings across all samples and printing conditions. Each of the 54 samples has been selected to mitigate potential experimental variability. Statistical analysis was employed to systematically demonstrate the influence of various printing regimes on surface roughness characteristics while accounting for potential confounding factors. Additionally, cross-validation was adopted with a comparison against established literature values to verify the reliability of the obtained results [57,58,59,60]. These methodological approaches collectively bolster the credibility of the findings, facilitating the interpretation of the influence (or lack thereof) of printing parameters on surface roughness in additive manufacturing processes.

### 4.3. Determination of Optimal Parameter for Pore Recognition

Figure 12 is the representation of manually detected pores in yellow, automatically detected pores in magenta, and the intersection of both in red depending on the patch size of automated pore recognition. The objective was to maximize the metric of the ratio of the area covered by red regions (representing true positives or TPs) to that covered by yellow regions. This was achieved by adjusting the size parameter of the Haar-like features representing the region with pores, which was defined as the edge length of the square

As for selecting the optimal size of Haar patches for pore recognition on monochromatic .tif images sized 3840 × 2160, captured at 33× magnification from the surface microstructure of an SLM printing pattern, it was found that a patch size of 36 pixels yielded the best results. For the analysis presented in Figure 12, the patch sizes of 30 and 36 pixels showed significantly higher recognition success rates. At patch sizes below 30 pixels, it was observed that nearly the entire image was covered with recognized squares, which rendered this approach impractical due to excessive overlap and reduced specificity. To ensure the robustness of findings, experiments were extended to include a range of patch sizes between 32 and 36 pixels. These assessments were conducted on the same 10 images for each patch size to maintain consistency. The results indicated that the optimal patch size remained within the range of 32 to 36 pixels, confirming that a patch size of 36 pixels was not an isolated peak but part of a consistent trend across different images, and a patch size of 36 pixels provides a balanced trade-off between recognition success and practical applicability, avoiding the excessive coverage observed with smaller patch sizes.

According to the suggested metric, the recognition success rate reached 82% with this patch size. However, it is important to note the potential limitations associated with this approach. On one hand, the accuracy of manual labeling [61] may vary, as human annotators might inadvertently include surrounding areas along with the pores or overlook some pores in the image. In contrast, an algorithm utilizing such a patch size can potentially adapt better to these variations.

The average percentage of the surface area containing detected pores for microstructure samples printed with laser parameters 175_80_80 was found to be 5.2%, while for 225_120_120 it was 4.2%, and for 275_160_160, it was 3.8%.

The SEM image in Figure 13 illustrates the surface of a sample produced using SLM with printing parameters configured to 225_120_120. Notably, this image showcases the automated detection of pores, providing insights into the sample’s microstructural characteristics.

## 5. Discussion

The study investigated water permeability in thermal pipes produced via SLM, focusing on the average percentage of pore area on the surface across various printing regimes.

The findings regarding water permeability underscore the complex interplay of factors influencing the integrity of SLM-produced thermal pipes. Despite variations in printing parameters such as laser power, scan speed, and layer thickness, the study did not identify a straightforward relationship between these variables and surface roughness, which in turn affects water permeability. This suggests that other factors, such as powder characteristics, melt pool dynamics, and post-processing techniques, may play significant roles in determining the porosity and surface quality of SLM-produced components.

Limitations of the study include the focus primarily on thermal pipes’ surface characteristics and the need for further investigation into the internal porosity distribution within the pipes. For instance, advanced imaging and analysis methods such as 3D X-ray computed tomography (CT) [62,63] or electron tomography [63,64] are required to precisely quantify and classify pore structures. These techniques can provide detailed insights into pore size distributions, connectivity, and their impact on water permeability. Additionally, it is important to explore different alloy compositions tailored for SLM processes to optimize material properties with enhanced fluidity [65,66], reduced susceptibility to oxidation, or improved microstructural stability under thermal conditions, aiming to mitigate pore formation [67] and improve overall the mechanical strength. Moreover, developing and refining post-processing strategies to minimize residual stresses and porosity via techniques such as hot isostatic pressing [68] (HIP), stress relief annealing [69], or surface treatments like chemical polishing or coatings [70] could be evaluated for their effectiveness in assessing surface quality and controlling permeability. Additionally, the utilization of computational tools to simulate the SLM process and predict microstructural evolution [71], including pore formation mechanisms and validation models against experimental data, lead to optimizing process parameters and predicting the impact of design modifications on water permeability and mechanical performance. Finally, investigation of multi-material or hybrid manufacturing approaches [72] combining SLM with other additive or subtractive techniques could help to explore how integrating different materials or functional layers can enhance thermal conductivity, reduce porosity, and improve resistance to fluid permeation in thermal pipes. So, while the average pore area percentage offers important observations, a more detailed analysis correlating specific pore sizes and distributions with printing parameters could enhance comprehension.

Square Haar-like patches have a fixed aspect ratio of 1:1, which can be limiting when pores or other features have irregular or elongated shapes. This fixed geometry may not effectively capture the anisotropy of certain features, leading to reduced recognition accuracy. Square patches [73] are equally sensitive to all edge orientations. However, pores may have preferred orientations due to the nature of the material processing. This isotropic sensitivity can result in a higher false positive rate when detecting features that are predominantly aligned in specific directions. While Haar features are computationally efficient, the use of only square patches can limit the complexity and richness of the features that can be detected. More complex shapes might be better suited to represent the variability of pores, but squares limit this potential diversity.

Future directions for research could explore advanced characterization techniques to precisely quantify and classify pore structures within SLM-produced pipes. Furthermore, investigating alternative alloy compositions or optimizing post-processing methods may offer insights into minimizing porosity and improving the overall performance of thermal pipes in practical applications. Moreover, adaptive Haar-like patches [49] that can change shape and size based on the local feature context represent a sophisticated approach. Machine learning algorithms can be trained to dynamically adjust patch shapes, providing the best fit for the detected features and thus enhancing recognition accuracy.

The surface roughness in SLM parts is well documented in the scientific literature. For example, Leon et al. [74] investigate the impact of surface roughness on the corrosion resistance and fatigue behavior of AlSi10Mg alloy produced via SLM. By comparing polished and unpolished SLM samples, the research reveals that polishing enhances both corrosion resistance and fatigue life. The increased surface roughness of unpolished SLM samples, attributed to inherent defects from the SLM process, leads to reduced corrosion resistance and fatigue endurance.

SLM enables the production of robust components at a low cost. However, inconsistent surface finishes pose a challenge to its widespread adoption. Leary [42] presents experimental findings on the surface roughness of SLM-manufactured titanium alloy Ti64, which is critical for structural and medical applications.

Majeed [75] explored the optimization of processing parameters for achieving high-quality products through the SLM of AlSi10Mg alloy. By employing various levels of scan speed, laser power, and overlap rate, they aimed to determine the optimal settings. Utilizing a 3-factor, 3-level full factorial Design of Experiments (DoE) with 27 tests, they collected experimental data. Subsequently, analysis of variance was employed to optimize the process parameters and achieve the best surface quality for AlSi10Mg alloy products manufactured via SLM.

Praneeth et al. [12] optimized parameters determined by Taguchi design; AlSi10Mg samples were produced and evaluated for mechanical properties. The results show that laser power and scan speed significantly impact hardness and surface roughness with maximum density achieved at 62.50 J/mm³ and maximum hardness of 125.6 HV.

Majeed [76] investigate the influence of processing parameters on the surface quality in SLM of AlSi10Mg alloy components. Laser power, scanning speed, overlap rate, and hatch distance are examined for their impact on surface roughness. Empirical exploration of laser power’s effect is followed by ANOVA to identify optimal power levels for minimal roughness. Regression analysis is then utilized to develop a mathematical model for optimizing the scan speed, overlap rate, and hatch distance to achieve minimal surface roughness. The study concludes that process parameters of 0.32 kW laser power, 0.60 m/s scan speed, 35% overlap rate, and 88.7 mm hatch distance yield optimal surface quality in the as-built condition. Additionally, a 17% reduction in average surface roughness is observed after solution heat treatment.

Yu et al. [77] explore remelting techniques in AlSi10Mg parts made via SLM. They assess surface roughness and porosity using various methods. Remelting significantly improves surface finish, reducing Ra values similarly in both directions. Pores, including spherical, irregular, and keyhole types, are reduced with remelting, aiding in their escape from melting pools. Same directional remelting is more effective at reducing porosity at edges due to differing distribution along melting tracks.

## 6. Conclusions

The analysis of water permeability across samples printed under varying conditions revealed notable trends. Specifically, we observed a linear increase in permeability with the increment of hatch distance, while an increase in laser power corresponded to a reduction in permeability. These trends suggest the essential role of printing parameters in determining the porosity and interconnectedness of the printed structures, which directly impact the fluid flow properties.

Surface roughness analysis reveals a weak correlation between printing parameters and surface roughness characteristics, while the parameters themselves that characterize surface roughness show strong correlations among each other. The automated detection of pores using ML techniques demonstrated promising results; there remains an active field of research for optimizing the recognition process. This includes exploring non-monochromatic images, images with different magnitudes, and the utilization of alternative Haar patch shapes. Nonetheless, the application of ML in pore recognition presents an exciting opportunity for streamlining quality control processes in additive manufacturing.

The study also sheds light on potential optimization strategies for enhancing the performance of SLM printing processes. By identifying optimal printing parameter combinations and refining surface analysis techniques, manufacturers can streamline production processes and improve the quality and consistency of printed thermal pipes.

## Figures and Tables

**Figure 1 sensors-24-04959-f001:**
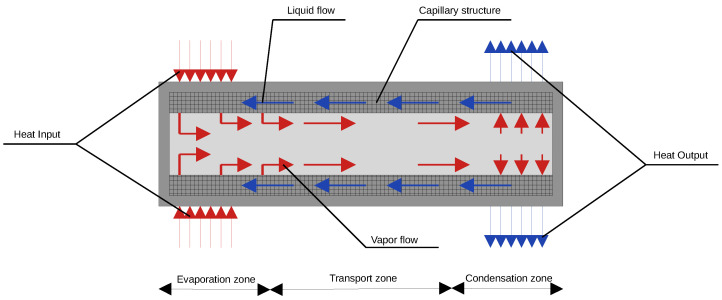
Schematic representation of an ideal one-dimensional thermal pipe model.

**Figure 2 sensors-24-04959-f002:**
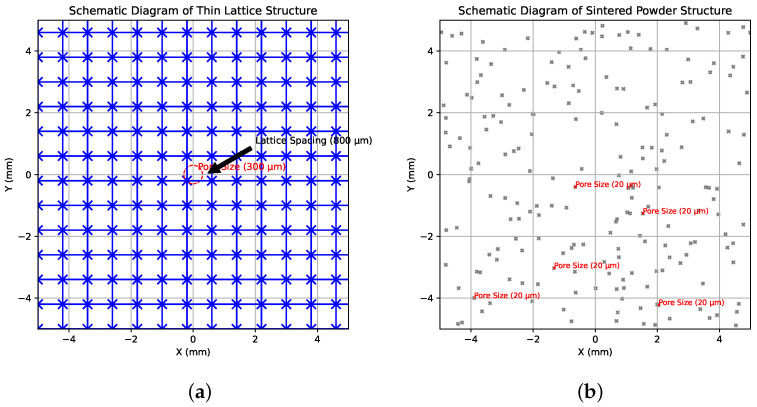
(**a**) Schematic diagram of thin lattice structure; (**b**) schematic diagram of sintered powder structure.

**Figure 3 sensors-24-04959-f003:**
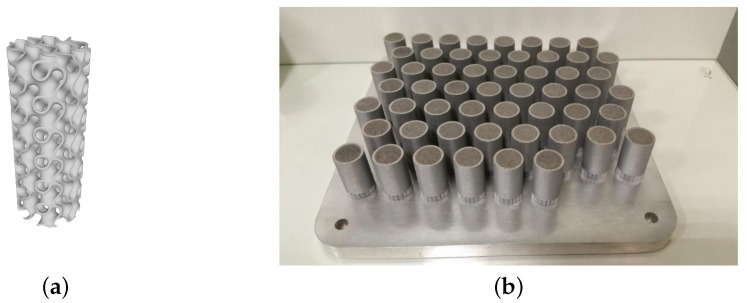
(**a**) Three-dimensional (3D) CAD inner model of sample; (**b**) platform with samples for water permeability determination: porous structure with dimensions of ⌀17 × 30 mm within a solid cylindrical shell. Alloy: AlSi10Mg.

**Figure 4 sensors-24-04959-f004:**
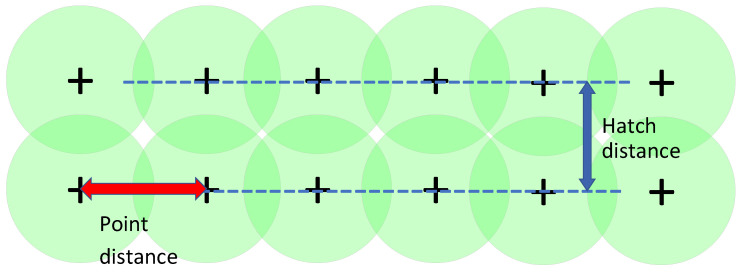
Key melting process parameters for controlling coherent porosity.

**Figure 5 sensors-24-04959-f005:**
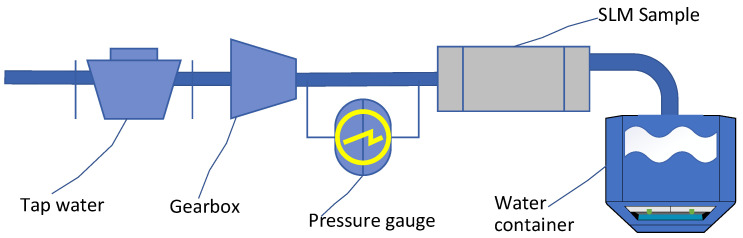
Experimental setup for permeability determination.

**Figure 6 sensors-24-04959-f006:**
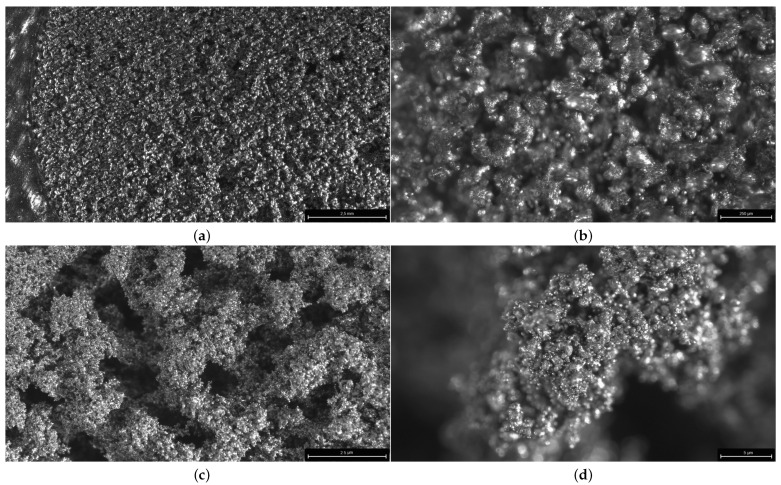
SEM images of the microstructure surface of the SLM part at different magnifications of the microscope: (**a**) 33×, (**b**) 55×, (**c**) 109×, and (**d**) 219×.

**Figure 7 sensors-24-04959-f007:**
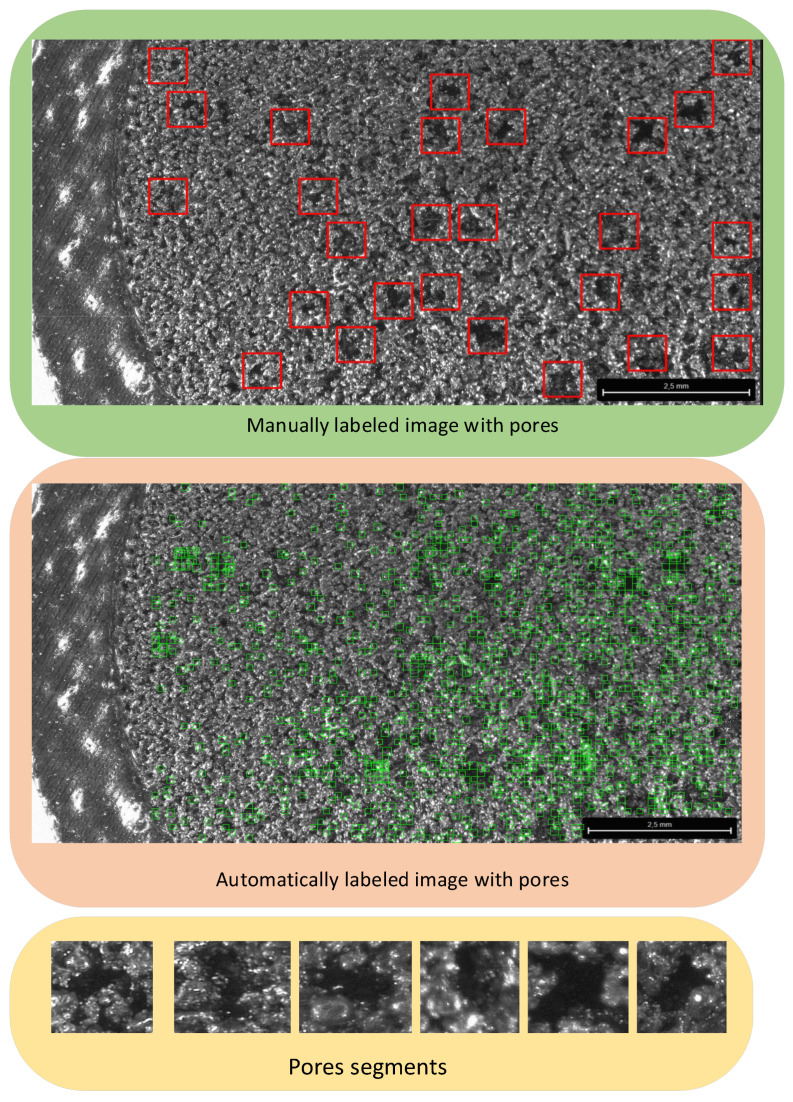
Schematic representation of the pore detection experiment.

**Figure 8 sensors-24-04959-f008:**
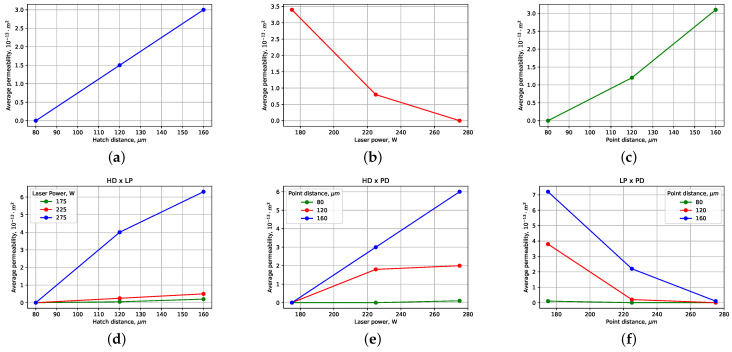
(**a**–**c**) Assessment of the influence of variable factors on permeability, line effects; (**d**–**f**) assessment of the influence of factors on permeability, pairwise interactions. Abbreviations used: LP—laser power, watts; HD—hatch distance, micrometers; PD—point distance, micrometers.

**Figure 9 sensors-24-04959-f009:**
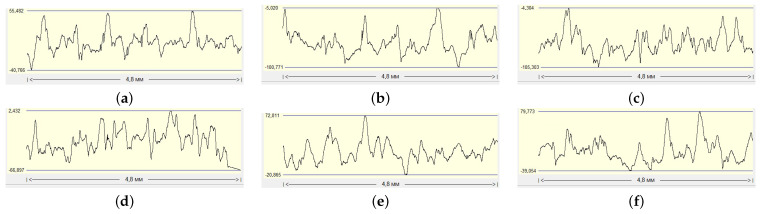
Examples of surface roughness profiles printed under different printing conditions: (**a**,**d**) laser power 175 W, hatch and point distances are 80 μm (175_80_80); (**b**,**e**) laser power 225 W, hatch and point distances are 120 μm (225_120_120); (**c**,**f**) laser power 275 W, hatch and point distances are 160 μm (275_160_160). For each printing regime, there are 18 samples.

**Figure 10 sensors-24-04959-f010:**
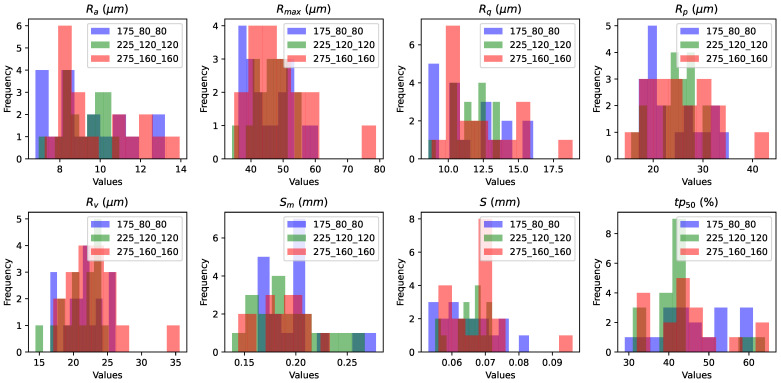
Histograms depicting the distribution of surface roughness parameters based on printing regimes. For each printing regime, there are 18 samples.

**Figure 11 sensors-24-04959-f011:**
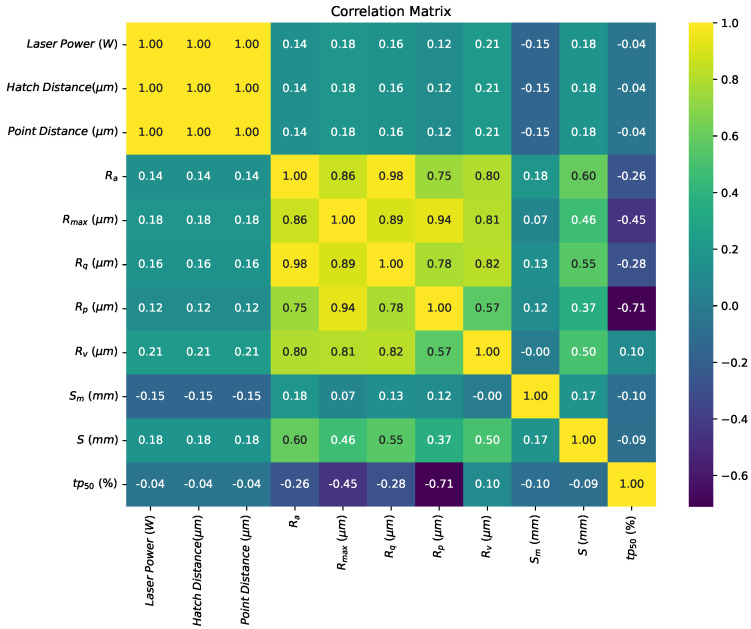
Correlation matrix showing the relationship between printing parameters and surface roughness parameters.

**Figure 12 sensors-24-04959-f012:**
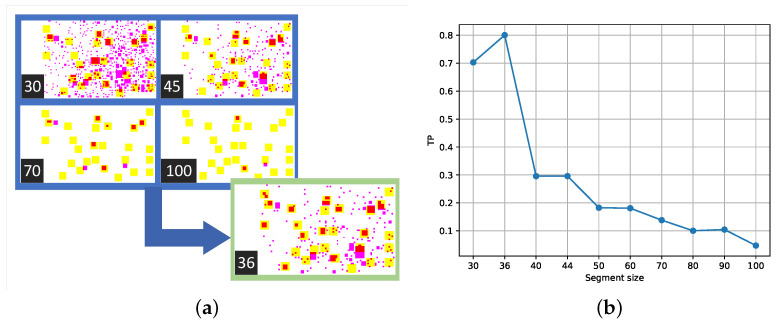
(**a**) Overlays pictures depending on Haar’s patch square image size for SEM image of SLM detail from Figure 7; (**b**) TP ration for this image depending on Haar’s patch square image size.

**Figure 13 sensors-24-04959-f013:**
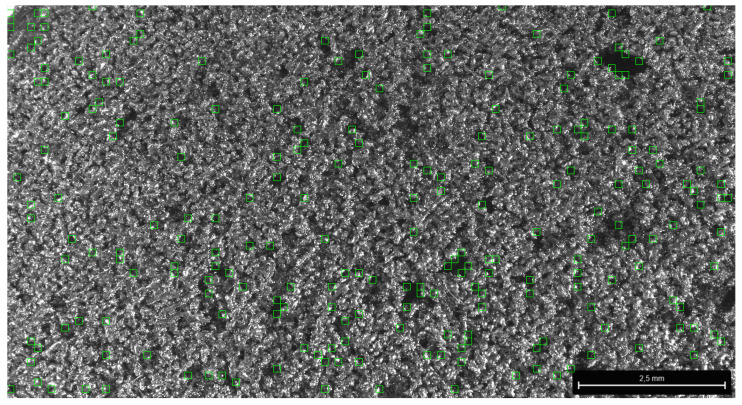
SEM image of the surface microstructure of the sample printed with SLM parameters 225_120_120, with automatically detected pores.

**Table 1 sensors-24-04959-t001:** Summary of studies on additive manufacturing applications in heat pipes and related structures.

Reference	Focus	Process	Material	Type of the Device
Gupta et al. [24]	Heat pipe operation principle, heat transfer via evaporation and condensation	Operation of heat pipes	Wick surface, vapor–liquid equilibrium	Heat pipe
Chen et al. [29]	Optimization of printing parameters for oscillating heat pipes	selective laser melting (SLM)	SUS316L	Oscillating heat pipe
Thompson et al. [30]	Creation of Ti-6Al-4V flat-plate oscillating heat pipe with closed-loop mini-channel	SLM, venting holes, SEM analysis	Ti-6Al-4V	Flat-plate oscillating heat pipe
Hu et al. [31]	Study of pore size effects on loop heat pipe (LHP) performance using 3D-printed wicks	3D printing of wicks, LHP operation	Stainless steel	Loop heat pipe
Esarte et al. [26]	Optimization of primary wicks in loop heat pipes (LHPs) via SLM	SLM, fluid dynamics in LHPs	Various metals	Loop heat pipe
Liu et al. [32]	Analysis of gas fluidity in SLM-fabricated cellular structures	SLM, porosity and permeability measurement	17-4 PH stainless steel, Inconel 718, Ti-6Al-4V	Cellular structures
Jafari et al. [33]	Investigation of thermal conductivity and wicking properties of SLM-fabricated stainless steel 316L porous structures	SLM, thermal conductivity, wicking properties	Stainless steel 316L	Porous structures
Monroe [34]	Study of Ti-6Al-4V flat-plate oscillating heat pipe via SLM	SLM, de-powdering method, SEM analysis	Ti-6Al-4V	Flat-plate oscillating heat pipe
Xie et al. [19]	Study on how laser power affects porosity in ZK60 magnesium alloys	SLM, porosity analysis	ZK60 magnesium alloys	Porous structures
Cerezo et al. [17]	Examination of fatigue behavior in additive manufactured samples with focus on pore presence	Additive manufacturing, fatigue testing	Various materials	Additive manufactured samples
Ye et al. [35]	Design and mechanical properties of titanium alloy porous scaffolds with TPMS designs	Selective laser melting (SLM), mechanical testing	Titanium alloy	Porous scaffolds

**Table 2 sensors-24-04959-t002:** Laser printing parameters.

Factor	Level at			Number of Samples
	−1	0	+1	
Laser Power, W	175	225	275	18
Hatch Distance, μm	80	120	160	18
Point Distance, μm	80	120	160	18

## Data Availability

Data are contained within the article.

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
