# Peer review of "Controlled Porosity of Selective Laser Melting-Produced Thermal Pipes: Experimental Analysis and Machine Learning Approach for Pore Recognition on Pipes Surfaces"

_sensors, 2024, doi:10.3390/s24154959_

Round 1

Reviewer 1 Report

Comments and Suggestions for Authors

The authors have provided a very interesting manuscript on controlling the porosity and permeability levels of selective laser melting produced samples of an AlSiMg alloy.  The paper is well written, but a few (likely minor) issues could be further clarified.

1. In the materials and methods section, please provide details of the powder used for the permeability sample printing.

2. How many samples printed and tested for each 3-factor condition for the permeability experiment?  This information should be added to the experimental writeup, or at least clarified.

3. Were the surface roughness analyses performed on the same samples as the permeability experiments?  Please clarify.  If so, you could move the printing description (with powder details) to the beginning of the materials and methods section.

4. How many of the 54 samples (line 158) scanned for surface roughness were from each 3-factor condition grouping?

5. For Figure 6, it would clarify the meaning further if the manual and automatically labeled images were side by side at the same magnification.   Then the pore segment image could be enlarged above these two images.

6. For the analyses performed in Figures in 8 and 9, it is unclear how many samples were utilized per 3-factor condition group?  This should be clarified, as insufficient values per group could influence the finding that the printing parameters did not influence surface roughness.

7. For the analysis performed in Figure 11, the 30 and 36 patch levels representing the smallest patches showed much higher levels of recognition success rate.  This would be intuitive, and not surprising.  However, it seems only two levels were assessed in this smaller patch size range?  Were any values below 30 patch size explored?  How local levels between 36 and 40?  In other words, how do we know a patch size of 36 was the true peak?  Was this assessment made on the same 10 images for each of the patch sizes?  These details need to be further clarified in the text write-up.

8. For the average percentage of surface area containing pores listed on lines 290 and 291 of the text, only two of the three 3-factor conditions are provided.  Please add the average value for the third group as well.

Author Response

Dear Reviewer1,

I hope this message finds you well.

Thank you for your insightful and constructive feedback on our manuscript titled. Your comments have been invaluable in enhancing the quality and clarity of our work.

Please find attached our detailed responses to each of your remarks. We have addressed all the points raised and made the necessary revisions to the manuscript accordingly.

We appreciate your time and effort in reviewing our paper and look forward to any further comments you might have.

Best regards, Ivan.

Reviewer 2 Report

Comments and Suggestions for Authors

Authors are requested to incorporate the suggestions mentioned in the review report

Comments on the Quality of English Language

Author Response

Dear Reviewer2,

I hope you are well. Thank you very much for your thorough and constructive review of our manuscrip. Your valuable feedback has greatly contributed to improving our work.

We have carefully addressed all your comments and made the necessary revisions. Please find our detailed responses in the attached file.

We appreciate your time and effort in reviewing our manuscript and welcome any further feedback you may have.

Sincerely, Ivan.

Reviewer 3 Report

Comments and Suggestions for Authors

Controlling the porosity of parts fabricated by SLM and using machine learning to recognize pores are promising areas to study. However, this article needs to present a deep enough investigation. Currently, many details are not clear and more quantitative conclusions are needed.

- Overall, it looks like that in this article, the study of the effect of parameters on water permeability, roughness, and pore recognition are three parts of independent works.

- It is suggested to draw schematic diagrams of two methods mentioned in Section 3.1 to tell what are the differences.

- As there are 54 samples mentioned in the article, a table is suggested to be given to summarize all samples and corresponding sample numbers and parameter sets for better understanding.

- For Figure 2a, what does this 3D part mean? Is it a part of the sample shown in Figure 2b? The geometric information of the sample printed by SLM needs to be clearly illustrated.

- Figure 5a and Figure 5c with different magnifications have the same scale bar length?

- Is it sure that images in Figure 5 taken by Leica S9 D are “SEM” images, especially Figure 5d it looks like an optical image?

- What do the terms like “printing regime” and “printing mode” mean?

- In Section 4.1, there are no images/tables to support the descriptions in the texts such as “large pores”, “smoother surface”, etc.

- The Section 5 discussion is listing literature, but it does not include related information regarding scientific analysis of this work. It is mentioned in Section 5 that "Surface roughness in SLM parts is well-documented in the scientific literature." How will the listed literature be related to the current work regarding surface roughness? And how this work differs from the documented literature?

- A few typos need to be corrected.

Author Response

Dear Reviewer3,

I hope you are doing well. We greatly appreciate your thoughtful and thorough review of our manuscript titled. Your feedback has been instrumental in refining and improving our work.

Attached, you will find our detailed responses to your comments. We have carefully addressed each point and revised the manuscript to reflect these changes.

Thank you for your time and valuable insights. We look forward to any further feedback you may have.

Best regards, Ivan.

Round 2

Reviewer 3 Report

Comments and Suggestions for Authors

Authors confirm that the microscopic images are not SEM images, but in the revised manuscript the phrase "SEM images" have not been corrected. 

For the content in Lines 295-297, it was not asked to add definition of "large pore", but to show images to readers. I did not see any image to show the comparison between larger pores and smaller pores under different parameters, such as laser powers, and similarly, attentions should also be paid in other parts of Section 4.

Author Response

Dear reviewer,

Thanks for your suggestions. 

Comment 1: For the content in Lines 295-297, it was not asked to add definition of "large pore", but to show images to readers. I did not see any image to show the comparison between larger pores and smaller pores under different parameters, such as laser powers, and similarly, attentions should also be paid in other parts of Section 4.

Reply 1: In lines 296 and 298, we added links to photographs of large and small pores.